## Replication

 

# A two-lab direct replication attempt of Southgate, Senju and Csibra (2007)

D. Kampis[1], P. Kármán[2], G. Csibra[2,3], V. Southgate[1] and M. Hernik[4]

[1]Department of Psychology, University of Copenhagen, Copenhagen, Denmark
[2]Department of Cognitive Science, Central European University, Budapest, Hungary
[3]Department of Psychological Sciences, Birkbeck, University of London, London, UK
[4]Department of Psychology, UiT The Arctic University of Norway, Tromsø, Norway

DK, 0000-0003-1678-3874

**Subject Areas:**
psychology

**Keywords:**
anticipatory looking, action anticipation, false belief attribution, replication, eye-tracking

**Authors for correspondence:**
D. Kampis
e-mail: dk@psy.ku.dk
M. Hernik
e-mail: mikolaj.l.hernik@uit.no

The study by Southgate *et al.* (2007 *Psychol. Sci.* **18**, 587–592. (doi:10.1111/j.1467-9280.2007.01944.x)) has been widely cited as evidence for false-belief attribution in young children. Recent replication attempts of this paradigm have yielded mixed results: several studies did not replicate the original findings, raising doubts about the suitability of the paradigm to assess non-verbal action prediction and Theory of Mind. In a preregistered collaborative study including two of the original authors, we tested one hundred and sixty 24- to 26-month-olds across two locations using the original stimuli, procedure and analyses as closely as possible. We found no evidence for action anticipation: only roughly half of the infants looked to the location of an agent's impending action when action prediction did not require taking into account the agent's beliefs and a similar number when the agent held a false-belief. These results and other non-replications suggest that this paradigm does not reliably elicit action prediction and thus cannot assess false-belief understanding in 2-year-olds. While the present results do not support any claim regarding the presence or absence of Theory of Mind in infants, we conclude that an important piece of evidence that has to date supported arguments for the existence of this competence can no longer serve that function.

## 1. Introduction

Motivated by the seminal study of Onishi & Baillargeon [1], which reported an understanding of false beliefs in 15-month-old infants using a violation-of-expectation paradigm, Southgate *et al.* [2] designed an eye-tracking study to test whether 25-month-old infants could anticipate actions on the basis of attributed false beliefs. The paradigm included two conditions which, together,

controlled for the possibility that infants might correctly predict without considering the other's belief. Data were reported from 20 infants, 10 in each condition, and the majority of infants appeared to correctly anticipate the agent's action in accord with her false-belief. Since publication, this study has been used as one of several key pieces of evidence for the existence of cognitive mechanisms that track others' beliefs in infancy [3,4].

The design of the study relied on anticipatory looking. It adopted the assumption that infants would be motivated to spontaneously anticipate through which of two windows an agent would reach for a ball. To elicit spontaneous anticipation, the study included two familiarization trials. The first showed an agent watching as a puppet placed a ball in one of two boxes, each located beneath one of two windows. Then, a light and sound cue were emitted simultaneously, after which the agent reached through the window above the box where she had seen the ball. Infants then received a second familiarization trial in which the agent observed the ball placed in the other box. If infants are motivated to predict which window the agent would open, the light and sound cues on this second familiarization trial should prompt them to look to the window above the box containing the ball before the agent would reach through the window. As the paradigm contained no 'task', and different participants may be motivated to attend to different aspects of the scene, only those infants who made a correct anticipation on this second familiarization trial were included in the analyses of anticipatory looking in the subsequent test trial. The reason for this criterion was that in order to interpret the data from the false-belief trial, evidence was needed for each infant that they were motivated and able to make a correct action prediction.

Due to the ease of eye-tracking with preverbal children and its ostensible immunity to factors that might vary across different laboratories, this paradigm has been employed extensively across different ages, populations and species. In later studies, either the original stimuli, or versions of, have elicited belief-based action prediction in neurotypical adults and school-aged children, but not in adults and children diagnosed with autism spectrum disorder (ASD) [5–7]. In other reports, anticipatory looking revealed differences in belief-based action prediction between typically developing children and children at-risk for autism [8] and deaf children growing up with impoverished language exposure [9,10]. Versions with even younger infants revealed belief-based action anticipation in 18-month-olds [11,12]. Most recently, new versions of the paradigm have revealed belief-based action prediction in apes [13,14] and monkeys [15].

However, in recent years, there have also been a growing number of papers reporting non-replications of belief-based action anticipation in toddlers and children [16–19] and even adults [18,20] in addition to non-replications of other anticipatory looking false-belief tests [20,21] as well as non-verbal false-belief paradigms using other methods (for an overview, see [22]). These reports are troubling, especially given that the original results were central in contributing to the reassessment of perspective-taking and mental-state attribution abilities of young children that challenged the long-held consensus that these abilities emerge relatively late in development. The non-replications have also highlighted a more fundamental problem with the paradigm. Specifically, that the paradigm may not reliably elicit the behaviour—action prediction—on which measuring attributions of false belief depends [22].

Nevertheless, only one of the reported non-replications was, strictly speaking, a direct replication of the original study. Specifically, one of the studies (Experiment 2) reported in Grosse Wiesmann *et al.* [17] was the only one that used the same number of familiarization and test trials, employed the original stimuli, tested approximately the same age group (24–27-month-olds, instead of the original 24–26 months age range), and was not run in conjunction with multiple anticipatory looking paradigms. However, the main analyses they reported did not include the original criteria of correct anticipation in the second familiarization trial.

In the current preregistered study, we aimed to carry out a direct replication of the Southgate, Senju & Csibra [2] study, following the original methods and analyses as closely as possible. The study was conducted with one hundred and sixty 25-month-old infants in collaboration across two locations: Copenhagen and Budapest.

## 2. Methods

The study closely followed the design of the original paradigm, with the exception that (i) more participants were recruited for the sample, (ii) the experiment was performed at two different locations (both different from the original), (iii) the first look was operationalized as fixation on either a window or a box to approximate the original manual coding of saccades to windows (see Dependent measures for details), (iv) the sound of a bell was used while the agent was turning away, (v) areas of interest (AOIs) for all gaze

analyses and the periods for inclusion criteria based on infants attending to the critical events were specified, (vi) a different brand of eye tracker was used, and (vii) the data-processing pipeline was implemented differently: in the original saccades and fixations were coded manually from a graphic representation of gaze generated in Clearview software, the present study used numerical data on saccades and fixations generated by the Eyelink online parser (see Dependent measures below and electronic supplementary material). The details of the data-processing pipeline and the analyses were preregistered (see https://aspredicted.org/sn78y.pdf), and all materials are available at https://osf.io/86wq2/.

## 2.1. Participants

In the original study, 36 infants were tested, and 20 of them were included in the analysis. In this replication study, we aimed for at least 40 infants who would provide data in the test trials after exclusions. As a pilot study indicated a high attrition rate, we decided to fix the full sample of infants at the number that would produce such a sample even if the exclusion rate was as high as 75%. We thus set the sample size at 40 infants tested in each condition at each site (i.e. 160 infants total prior to applying the exclusion criteria).

All exclusion criteria were preregistered, and were matched as closely as possible to the original study (for details see preregistration and electronic supplementary material). In the original, participants were excluded if they did not correctly anticipate the opening of the right-hand window on the second familiarization trial, looked away at the crucial moment on the test trial, did not look at either window on the test trial, or the eyetracker could not be calibrated for them. We specified analogous criteria in more detail in our preregistration. We intended to ensure that infants in the included sample were attending to key events in familiarization trials and thus established a criterion for minimum looking during predefined events (see below). Note that the original paper did not report any exclusions resulting specifically from infants looking away during these events. In the supplement, we also analysed the data including infants who did not meet our criteria of attending to key events in familiarization and test trials.

Consequently, we excluded participants if: (i) there were procedural errors, (ii) the infant was looking at the scene for less than 2 cumulative seconds during the key periods in familiarization, (iii) the infant was looking away during the key periods (for specifications of key periods see electronic supplementary material) at test for more than 1.5 consecutive seconds, (iv) the infant made no fixation to a window or box within 2750 ms from cue onset in the test trial, or (v) the infant was not looking at the scene during the cue in test or in the second familiarization trial. Infants were excluded from the main analyses if they did not anticipate correctly on the second familiarization trial.

The study was approved by the Ethics Committee of the University of Copenhagen Department of Psychology and the United Ethical Review Committee for Research in Psychology (EPKEB) in Hungary for the two sites, respectively, and parents signed an informed consent prior to participation.

In the current study, 160 two-year-old infants participated (mean age: 24 months and 24 days; range: 721–808 days; 95 males, 65 females; 124 monolinguals, 35 multilinguals (15% of the time is exposed to at least one other language besides the primary language), participants in the sample were exposed to: Danish, Hungarian, Icelandic, Swedish, English, Faroese, Dutch, Arabic, French, Romanian, Spanish, Russian, Ukrainian, Portuguese, Italian, German, Chinese, 1 N/A). Eighty infants were tested at the University of Copenhagen (mean age: 24 months and 22 days; range: 721–808 days; 48 males, 32 females; 58 monolinguals, 21 multilinguals, 1 N/A) and 80 infants were tested at the Central European University in Budapest (mean age: 24 months and 25 days; range: 728–788 days; 47 males, 33 females; 66 monolinguals, 14 multilinguals).

Of the 80 participants tested in Copenhagen, 25 infants were included in the main analysis (mean age: 24 months and 20 days, range: 735–784 days; 15 males, 10 females; 16 monolinguals, 8 multilinguals, 1 N/A), and 55 infants were excluded for the following reasons: procedural errors (6), looking at the scene for less than 2 cumulative seconds during the key periods in familiarization (7), looking away during the key periods at test for more than 1.5 consecutive seconds (26), not looking at the scene during the cue in second familiarization trial (1) or not anticipating correctly on the second familiarization trial (15).

Of the 80 participants tested in Budapest, 24 infants were included in the main analysis (mean age: 24 months and 22 days, range: 728–788 days; 10 males, 14 females; 21 monolinguals, 3 multilinguals), and 56 infants were excluded for the following reasons: procedural errors (5), looking at the scene for less than 2 cumulative seconds during the key periods in familiarization (1), looking away during the key periods at test for more than 1.5 consecutive seconds (20), no fixation to a window or box within 2750 ms from cue onset in the test trial (4) or not anticipating correctly on the second familiarization trial (26)

Altogether, 49 participants who passed all the inclusion criteria were included in the main analyses (mean age: 24 months and 21 days, range: 728–788; 25 males, 24 females; 37 monolinguals, 11

multilinguals, 1 N/A). Out of these 49 participants, 31 infants remained in the false belief 1 (FB1) condition and 18 in the false belief 2 (FB2) condition.

The final sample for the main analysis was approximately 2.5 times the original sample, as recommended for replication studies [23].

## 2.2. Apparatus

Eyelink1000 Plus eye trackers were used to collect gaze data in both locations. In order for the eye tracker to be able to follow the participants' gaze, infants wore a target sticker on their forehead or cheek. The eye tracker tracked the movement of the participant's right eye. The stimuli were presented on a 17 inch monitor (Copenhagen) or on a 24 inch monitor (Budapest) via a PC running Eyelink's Experiment Builder software (SR Research). In the original study, a Tobii 1750 eye tracker with an integrated 17 inch monitor was used, while the stimuli were presented using the Clearview AVI presentation software.

## 2.3. Procedure

When the child entered the lab, an animation with rotating colourful objects was played on the screen, accompanied by music. This animation was presented during the seating and preparation. Infants were seated on their parents' lap, 60 cm from the screen. The parents either wore opaque glasses or were asked to close their eyes during the entire study. The whole experiment was video recorded for each participant. (Videos were not recorded for three participants: two from Copenhagen, one from Budapest. However, as exclusions and all analyses were performed on eyetracking data, these infants were included in our sample). After the initial animation, Eyelink's 5-point calibration was carried out for each infant. After successful calibration, the experiment began.

We used the original stimuli [2] with one modification. In the test trials, when the agent turned away, instead of the sound of a ringing phone, the sound of a bell was played. We reasoned that infants in 2019 may not be familiar with the phone sound used in the 2007 version. As in the original study, there were no attention getters between trials.

Each infant was presented with three video clips: two familiarization trials and one test trial. In all three trials, a female agent appeared behind a panel containing two windows, below which were two opaque boxes. All three trials began with a bear puppet coming into view from the bottom of the screen with a colourful ball and then placing this ball into one of the boxes. In the first familiarization trial, the puppet hid the ball in the left-hand box, while in the second familiarization the puppet hid it in the right-hand box. After hiding the ball, the puppet left the scene. After this, a sound and light cue (the latter on the windows) were played for 1 s, and following a 1750 ms delay, the agent reached through the left-hand window. In the first familiarization she opened the box lid and retrieved the ball. In the second familiarization trial, she reached through the right window and the trial ended when she made contact with the box. As in the original study, infants then saw one of two test trials, depending on condition.

In the FB1 condition, infants saw the puppet put the ball first into the left-hand box, then take it out and place it into the right-hand box and leave. Then the sound of a bell was played and the agent turned away as if attending to the source of the sound. While she was turned away, the puppet reappeared, took the ball out of the right-hand box, and left. The bell stopped ringing when the puppet left and the agent turned back towards the child.

In the FB2 condition, infants saw the puppet put the ball into the left-hand box and leave. At this point, the bell started ringing and the agent turned away. While she was facing away, the puppet reappeared, took the ball from the left-hand box, and transferred it into the right-hand box. Then, the puppet retrieved the ball from the right-hand box and left with it. When the puppet left the scene, the bell stopped ringing and the agent turned back.

In both test trials, after the agent turned back towards the child, the same sound and light cue were presented as in familiarization, followed by an approximately 5 s period when the movie froze. Unlike in familiarization trials, the agent remained stationary and did not reach through either window.

After the lab session, parents filled out a basic demographic questionnaire (adapted from the ManyBabies1 project: https://osf.io/56fze/).

## 2.4. Dependent measures

We defined two different pairs of AOIs (see electronic supplementary material for details). The window + box AOIs included both the window and the box on one side of the screen, while the window-only AOIs were defined only around the window on each side.

The period of interest was defined as the 2750 ms interval starting from the onset of the acoustic and visual cues that indicated the agent's impending action. In the original paper, the period of interest was described as 1750 ms from the cue onset and it was said to correspond to the time-lag between the cue onset and window opening in familiarization. In fact, this time-lag lasts 2750 ms. After inspecting the original stimuli and data as well as consulting among the authors, we concluded that the length of 1750 ms was reported incorrectly in the original methods and we corrected it to 2750 ms for the current replication. We do, however, also report analyses for the period of interest lasting 1750 ms from the cue onset (see electronic supplementary material).

For measurement, we used two dependent variables. The primary dependent variable was binary: it expressed whether the first window + box AOI that the participant fixated within the time window of interest was on the correct side. Window + box AOIs were used in order to approximate the operationalization in the original study, where the first *saccade* to a window was coded from movies showing graphical representation of gaze generated with Tobii's Clearview software. Given that saccades were coded from video replay of gaze location, it is possible that the original coding could have included saccades leading through the window towards the box below. We additionally report first look to a window-only AOI in the electronic supplementary material.

The secondary dependent variable was a differential looking score (DLS) calculated for each participant as cumulative fixation time to correct window-only AOI divided by the sum of cumulative fixation times to the two window-only AOIs during the time window of interest. For this analysis, we analysed fixations to the window only, as this approximates well the coding of looking times used in the original study. Analysing the DLS measure is preferred to comparing looking durations to the two AOIs because looking durations are not independent from each other. Nevertheless, in the electronic supplementary material, we also report the same statistical analyses on looking durations as those reported in the original paper.

## 2.5. Analyses

All methods and analyses were preregistered prior to the start of data collection (https://aspredicted. org/sn78y.pdf). Subsequently, the Stage 1 version of the manuscript, not including results and discussion, received results-blind in-principle acceptance (IPA) at *Royal Society Open Science*. As per journal policies, following IPA the accepted Stage 1 version was uploaded as registration to the OSF (https://osf.io/bsf2r?view_only=0a90aeddc40543fb85f7ef8e8e8f53cf). The registration of Stage 1 manuscript was thus performed after data analysis, but the analyses were done as described in the original preregistration (see above).

# 3. Results

From the full sample, 70 infants were excluded for procedural errors, inattention or lack of anticipatory fixations at test. Of the 90 remaining infants, 49 (54%) anticipated correctly in the second familiarization trial. This is a somewhat lower proportion than what was reported in the original study (20 of 31, 65%).

Our primary dependent measure was whether infants anticipated correctly (i.e. whether after the cue onset, the first window or box on which they fixated was on the side where the agent should have expected to find the ball, given her false belief). Twenty-two of the 49 infants anticipated correctly, a proportion that did not differ from chance, $p = 0.568$, two-tailed exact binomial test. Correct anticipations were found in 15 of 31 infants in the FB1 Condition and in 7 of 18 infants in the FB2 condition. The proportion of correct responses did not differ significantly between the conditions, $p = 0.565$, two-tailed Fisher's exact test.

Our secondary dependent variable was the DLS score calculated from the time infants looked at one or the other window-only AOI. The mean DLS score was 0.47, 95%CI [0.36 0.57]. A score above 0.5 would have indicated longer looking to the correct AOI. A one-sample *t*-test on the DLS scores against chance level of 0.5 indicated that infants did not look longer at the correct AOI than the incorrect AOI, $t_{47} = -0.63$, $p = 0.532$. One infant was not included in this analysis because they fixated first a box but never fixated a window. An independent-samples *t*-test with condition (FB1 versus FB2) as a grouping variable showed that DLS was higher in the FB1 ($M = 0.54$) than in the FB2 ($M = 0.33$) condition, $t_{35.76} = 2.04$, $p = 0.049$, 95% $CI_{diff}$ [0.001 0.41]. This effect is illustrated in figure 1 and was primarily due to the fact that infants tended to look longer at the right window, which was correct in FB1 and incorrect in FB2, thus leading to a condition effect. DLS was only numerically higher than chance level of 0.5 in FB1, $t_{30} = 0.64$, $p = 0.527$,

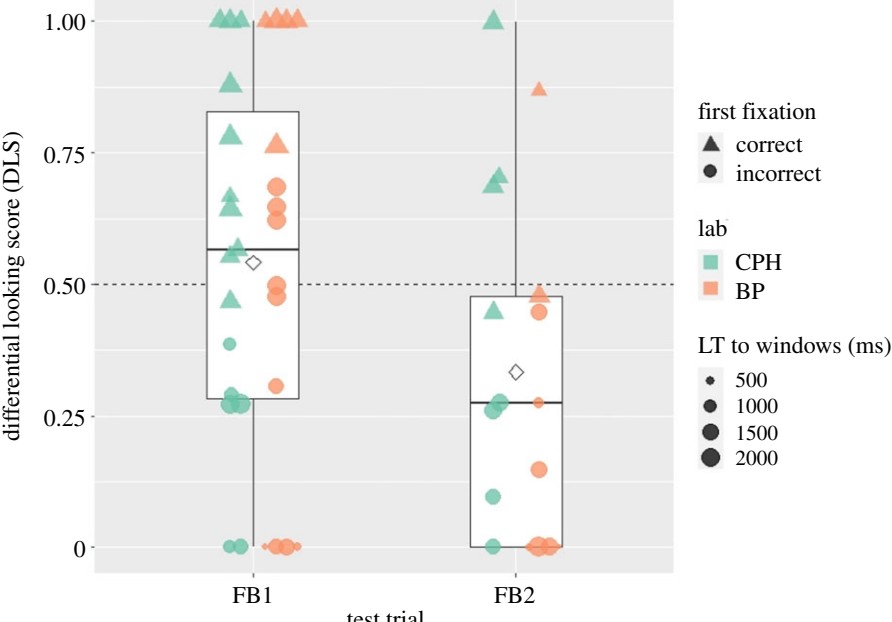

**Figure 1.** Differential looking score (DLS) in the two conditions in the group of infants who passed all inclusion criteria and correctly anticipated in the second familiarization trial. Note that one infant did not provide a DLS score, as they did not fixate on a window. Boxplots show median, first and third quartile, minimum and maximum values. White diamonds indicate condition means. Individual data points are plotted per lab: left side of each box plot—Copenhagen (CPH), right side—Budapest (BP). Shapes indicate whether the first fixated window or box was on the correct (triangle) or incorrect side (disc). The size of individual data points scale with the amount of looking time (LT) to both windows contributed by each participant.

and numerically lower than chance in FB2, $t_{16} = 2.11$, $p = 0.051$; both tests are one-sample $t$-tests on the DLS scores against chance level of 0.5.

The above analyses only included infants who looked first to the correct window or box on the second familiarization trial. We considered *post hoc* that the difference in DLS between the two conditions could be driven either by a general right-side bias (because it was the location of the ball in the last familiarization trial, or because it was the last location of the ball in test) or by an individual side bias, i.e. looking to the same side in two consecutive trials (last familiarization and test trial). The results of the included infants alone could not differentiate between these alternatives. In a follow-up analysis performed on all infants who passed all other exclusion criteria and in the second familiarization trial produced either the 'correct' ($N = 49$, included infants) or the 'incorrect' ($N = 41$, excluded infants) first fixation to window + box AOI, we asked whether either of these tendencies might be visible. We conducted an ANOVA with DLS as a dependent variable and condition (FB1 versus FB2) and inclusion (included versus excluded) as between-subject factors. As a DLS above 0.5 reflects longer looking to the correct AOI, it indicates longer looking to the right window in FB1 and to the left window in FB2. An overall bias to the right side, therefore, in this extended sample would manifest in a main effect of condition. On the other hand, a potential individual side bias to look to the same location in the second familiarization and in test trial should manifest in a main effect of inclusion, as exclusion and inclusion correspond to looking left and looking right in second familiarization, respectively.

This analysis resulted in a significant effect of inclusion, $F_{1,85} = 4.77$, $p = 0.032$, as the included infants had overall higher DLS than the excluded ones. In addition, there was also a significant interaction between condition and inclusion, $F_{1,85} = 7.00$, $p = 0.010$, see figure 2. Thus overall, in the test trials, infants tended to look to the same side as where they fixated in a window + box AOI first in the preceding familiarization trial, and this pattern was more manifest in the FB2 than in the FB1 condition. This finding indicates an individual side bias. Since this effect is irrelevant for the purpose of the study, we will not discuss it further.

In the electronic supplementary material, we report results of the following preregistered additional analyses: (i) an ANOVA with duration of looking to the correct versus incorrect window AOIs as within-subjects factor, and condition (FB1 versus FB2) as between-subjects factor (as reported in the original

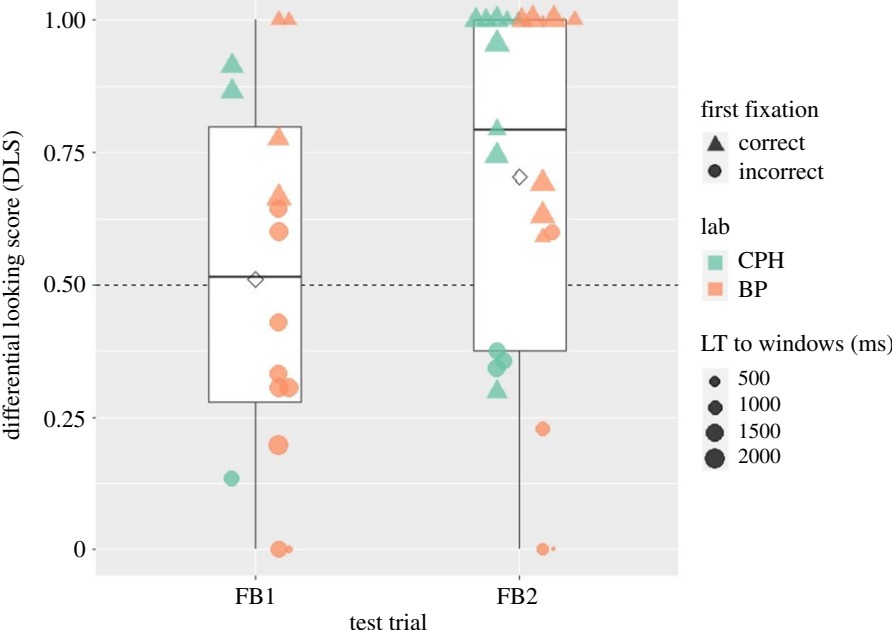

**Figure 2.** Differential looking score (DLS) in the two conditions in the group of infants who passed all inclusion criteria, except that they anticipated incorrectly in the second familiarization trial. Boxplots show median, first and third quartile, minimum and maximum values. White diamonds indicate condition means. Individual data points are plotted per lab: left side of each box plot—Copenhagen, right side—Budapest. Shapes indicate whether the first fixated window or box was on the correct (triangle) or incorrect side (disc). The size of individual data points scale with the amount of looking time (LT) to both windows contributed by each participant.

article); (ii) the main analyses without using the look-away criteria in familiarization and test trials (as they were not included in the original study); (iii) the main analyses with excluding those who did not fixate on head AOI before fixating on window + box AOI (as original paper found all participants in the final sample saccade to window from the head); and (iv) the main analyses with a period of interest of 1750 ms from stimulus onset (as this was the period reported in the original study). None of these additional analyses yielded substantially different results.

# 4. Discussion

This paradigm was originally intended to serve as a non-verbal means of assessing whether young children anticipate others' actions in a way that takes into account their false beliefs. A positive answer to this question would be considered evidence of false belief attribution. However, having sampled one hundred and sixty 25-month-old infants across two sites, we found no evidence for either action anticipation or belief-based action anticipation. Of the participants who were not excluded for other reasons, only 54% of infants correctly anticipated the impending action in the second familiarization trial, where anticipation did not require belief attribution. Furthermore, on the critical test trial, where an action could be anticipated by considering the agent's false belief, only 45% of infants anticipated correctly.

Thus, in the present study, we were unable to replicate the original finding of Southgate *et al*. [2]. This replication attempt, unlike most of the earlier ones (for a discussion see [22]), was very close to a direct replication (see Methods for the exceptions). Where we did deviate from the procedure as described in the original paper, the changes aimed at approximating (e.g. when specifying AOIs), improving (e.g. when predefining the critical periods infants should attend to) or clarifying (e.g. by correcting the description of the period of interest to 2750 ms from cue onset) the original methods, and we have no reason to suspect that these differences would systematically account for infants' failure to anticipate actions in the present study. Overall, we conclude that this paradigm and stimuli do not reliably elicit action anticipation. That is, the paradigm does not elicit the behaviour upon which it relies to assess the ability to attribute false beliefs, and therefore it does not provide evidence for this ability in 2-year-old children.

While some follow-up versions of the original paradigm reported high levels of correct anticipation on the second familiarization trial when no belief consideration was required [6,11,24], several other studies found that participants anticipated either at or even below chance (see [22] for a review). Why do children (and even adults, see [18]) not make correct anticipations on familiarization trials? Other paradigms, including those using anticipatory looking, show that infants can generate the kind of goal-based action predictions required to look to the correct location on the second familiarization trial ([25–29]; but see [30]). A possible answer to this question is that either this task does not motivate infants to predict others' actions, or that action prediction in this kind of event does not reliably elicit eye movements to the target location. An indication that participants may not produce saccade and fixation patterns that clearly reveal their predictions comes from a replication and extension of the original study. Wang & Leslie [31] included two false belief conditions; a low-demand condition in which the object is removed from the scene in order to reduce the so-called 'pull of the real' and a high-demand condition in which the object is transferred from the original location to the other location. While both young children and adults showed correct action prediction on the low-demand condition, neither group showed correct action anticipation on the high-demand condition. Assuming that adults should easily be able to make the correct prediction in the high-demand condition, the most likely explanation for their apparent failure is simply that actual prediction is not reflected in their saccade pattern.

A further way in which our findings differed from those of the original study is that in the current study there was a higher exclusion rate due to inattention (34% versus 6%). However, direct comparison of exclusion rates may not be appropriate given the differences in implementation of the exclusion criteria across the two studies. For example, in the original study, looking was coded from movies showing graphic representation of gaze generated with Tobii's Clearview software, whereas in the current study it was measured by exported fixation data. Given that we used the same stimuli as in the original study, it is possible that what was sufficient to motivate toddlers to attend to the stimuli in 2006 is different from what is sufficient to motivate them 14 years later. Moving forward, based on such considerations, new projects such as the multi-lab collaborative ManyBabies2 will present toddlers with more modern, newly devised stimuli that are designed to be more appealing [32].

Besides not providing evidence for goal-based action anticipation, our study also failed to reproduce the original finding according to which most of the 25-month-olds who appeared to anticipate during familiarization also did so when action prediction could have been made from attributed false beliefs. While in the original study infants were included based on the assumption that they had shown correct anticipation on the second familiarization trial, it should be acknowledged that what was interpreted as correct anticipatory looking may have been randomly distributed exploration or reaction evoked by the lighting up of the windows (20 of 31 infants appeared to anticipate correctly). Thus, it is possible that the original criteria did not select children who were genuinely predicting an action from the goal attributed to the agent. While the majority of those children did then look correctly again on test, the current failure to replicate this pattern with a larger sample indicates that they may have done so by chance. We, therefore, conclude that the effect reported by Southgate *et al*. [2] is not reliable and should not be used as evidence for false-belief attribution in children of this age.

Nevertheless, it is important to note here that this, and other replication failures of this paradigm, cannot be used as evidence for the opposite conclusion either. Rather than showing the absence of competence in belief attribution, the fact that infants did not anticipate actions even when they did not have to think about the other's belief renders these current null results uninterpretable. Other paradigms may also support the conclusion that infants and toddlers track others' attention and build expectations based on what others have or have not seen [4,33]. Similarly, our failure to replicate the original finding does not nullify the positive results obtained with versions of this paradigm in different populations, such as non-human primates. Thus, the current results should be taken as damning of this particular version of the paradigm with this particular age group, rather than the entire body of evidence for false belief understanding in infants or in other populations. Nevertheless, we conclude that an important piece of evidence that has to date supported the existence of early competence of mental-state attribution should no longer serve this function.

Ethics. The study was approved by the Ethics Committee of the University of Copenhagen Department of Psychology and the United Ethical Review Committee for Research in Psychology (EPKEB) in Hungary for the two sites respectively, and parents signed an informed consent prior to participation.

Data accessibility. The details of the data-processing pipeline and the analyses were preregistered (see https://aspredicted.org/sn78y.pdf), and all materials are available at https://osf.io/86wq2/.

The data are provided in electronic supplementary material [34].

Authors' contributions. D.K. and P.K. collected the data, M.H. performed the analysis, and D.K., P.K., G.C., V.S. and M.H. conceived and designed the study and wrote the paper.

Competing interests. We declare we have no competing interest.

Funding. We received no funding for this study.

Acknowledgements. The publication charges for this article have been funded by a grant from the publication fund of UiT The Arctic University of Norway.

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
