## [Peer Review File · Royal Society Open Science]

Review History

RSOS-210190.R0 (Original submission)

Review form: Reviewer 1

Do you have any ethical concerns with this paper?

No

Have you any concerns about statistical analyses in this paper?

No

Recommendation?

Accept with minor revision

Comments to the Author(s)

I highly appreciate the authors' approach to conduct a close direct replication of their own task with a sufficiently large sample. Although I am not convinced by the authors' argumentation that previous failed replications of their paradigm suffer from tremendous methodological deviations, more replication data -especially in the original age group- will contribute to get a clearer view on the bigger picture of early mindreading capacities.

I only have some minor comments for the authors:

- Introduction, second paragraph: “a light and sound cue was emitted simultaneously” I think it should read “...cue were emitted...”
- Introduction, fourth paragraph: Dörrenberg et al. (2018) tested 24-month-olds, this study should be listed behind toddlers, not children. Schuwerk et al. (2016) tested 8-year-olds with and without ASC – I guess the authors refer to a different study: Schuwerk T, Priewasser B, Sodian B, Perner J. 2018 The robustness and generalizability of findings on spontaneous false belief sensitivity: a replication attempt. *R.Soc.opensci.* 5:172273. <http://dx.doi.org/10.1098/rsos.172273>
- Introduction, fifth paragraph: The authors claim that there is only one published study meeting the criteria of a direct replication of the task reported by Southgate et al. in the 2007 paper. This statement seems to question the validity of all other previous replication studies. Yet, the authors do not go into detail for what reasons they discredit these studies. As far as I can see, many previous replication studies are at least close to direct replications with only minor deviations or used methods/age groups that resemble or lean on follow-up studies of this paradigm by the original authors. It is worth noting that also the current study differs in some ways from the original one (e.g., different dependent measure, data processing, exclusion criteria, ringing sound, etc.). To give readers the chance to make their own -unbiased- evaluation of the current situation, I would appreciate if the authors would provide some more information on how previous replication studies differed from the original study and explain why they think these differences would question their validity. For instance, some replication studies used a “four trial” familiarisation procedure (e.g., Dörrenberg et al. 2018, Kulke et al. 2018). Why should such a procedure have negative consequences on the results of these studies? In fact, the original authors themselves introduced the two extra familiarisation trials (that only show a short sequence of the agent reaching for a toy on the box) to the task in follow-up studies. Other replication studies used new stimulus material that they created in conformity with the original material (e.g., Kulke et al. 2018, Schuwerk et al. 2018). Why should only the original footage work in eliciting belief-based action anticipation? And so forth.

Review form: Reviewer 2

Do you have any ethical concerns with this paper?

No

Have you any concerns about statistical analyses in this paper?

No

Recommendation?

Accept with minor revision

Comments to the Author(s)

The study is begging to be done. Over the last several years, there has been considerable discussion regarding the original findings from Southgate et al. (2007), whether these effects are robust, and what this means for research on early false-belief understanding. Yet, as the current authors note, no one has undertaken a *direct* replication, which makes this entire debate somewhat difficult to interpret. This direct replication conducted across 2 labs with a more robust sample size will thus make a valuable and important contribution to the field.

The study plan mirrors the original study quite closely, with only minor changes related to the eye-tracking equipment currently available. The authors plan to replicate the original analyses, while also conducting additional analyses to examine differences across lab, etc.

The preregistration on aspredicted.org is quite clear. However, the manuscript itself could be clarified in several places to help the reader understand how the current study is similar to/different from the original study that was conducted. The original paper was quite brief and did not discuss some criteria (such as exclusion) in great detail, so it would be helpful to provide additional detail in the current manuscript. Note all of these suggestions are intended to clarify the eventual manuscript. In my opinion, they do not affect the quality of the replication study itself, as all of these issues are clearly addressed in the online study plan.

- It is briefly mentioned on pg 8 that the original study coded gaze manually from a graphic representation of gaze generated in Clearview. The present study will instead use saccades/fixations as defined by the Eyelink software. It would be helpful to explain this difference earlier so that it is clear why two different AOIs are defined.

- Similarly, it would be helpful to state clearly in the manuscript whether the exclusion criteria were the same as in the original study. This is discussed in the supplementary material and preregistration, but it is somewhat unclear in the manuscript.

- In the preregistration, it states that the plan is to test 40 infants in each condition at each site (i.e. 160 infants total prior to applying the exclusion criteria). The manuscript does not lay this out as clearly, so it seems as though only 40 infants will be tested total. Presumably this will be clarified in the final manuscript when actual data are reported.

- The manuscript notes a discrepancy in the original paper regarding the analysis window. However, the nature of this discrepancy is somewhat unclear. From the preregistration, it seems like the duration from cue onset to the agent's reach (in familiarization) was 2750ms and the duration from cue *offset* to reach was 1750ms. But it is not clear which of these two windows as actually analyzed in the original paper - are the analyses in the original paper on the 2750ms window or the 1750ms window? In the current study, the authors plan to analyze both, so whichever it was the current paper will provide a direct replication of the original analyses. But it would be quite helpful to the field to clarify which was originally used, given that others have (or are currently) undertaken conceptual replications of this work.

Decision letter (RSOS-210190.R0)

Dear Dr Kampis

On behalf of the Editors, I am pleased to inform you that your Manuscript RSOS-210190 entitled "A two-lab direct replication attempt of Southgate, Senju, & Csibra (2007)" deemed suitable for in-principle acceptance in Royal Society Open Science subject to minor revision in accordance with the referee and editor suggestions. Please find their comments at the end of this email.

The reviewers and handling editors have recommended publication, but also suggest some minor revisions to your manuscript. Therefore, I invite you to respond to the comments and revise your manuscript.

Please you submit the revised version of your manuscript within 7 days (i.e. by the 04-Mar-2021). If you do not think you will be able to meet this date please let me know immediately.

When submitting your revised manuscript, you will be able to respond to the comments made by the referees and upload a file "Response to Referees" in the "File Upload" step. You can use this to document any changes you make to the original manuscript. In order to expedite the processing of the revised manuscript, please be as specific as possible in your response to the referees.

Full author guidelines can be found here <https://royalsocietypublishing.org/rsos/replication-studies#AuthorsGuidance>.

Kind regards,
 Professor Chris Chambers
 Royal Society Open Science
openscience@royalsociety.org

on behalf of Professor Chris Chambers (Registered Reports Editor, Royal Society Open Science)
openscience@royalsociety.org

Associate Editor Comments to Author (Professor Chris Chambers):

Associate Editor: 1

Comments to the Author:

Two expert reviewers have now assessed the Stage 1 manuscript. The assessments are broadly very positive, with both reviewers judging that primary criteria #1 and #2 are met. Key issues to address include the framing of past research in the introduction and additional methodological detail to clarify ways in which the replication differs from (or aligns with) with the original study. Provided the authors are able to respond comprehensively to these points in a revision, in-principle acceptance should be forthcoming without requiring further in-depth Stage 1 review.

Reviewers' comments to Author:

Reviewer: 1

Comments to the Author(s)

I highly appreciate the authors' approach to conduct a close direct replication of their own task with a sufficiently large sample. Although I am not convinced by the authors' argumentation that previous failed replications of their paradigm suffer from tremendous methodological deviations, more replication data -especially in the original age group- will contribute to get a clearer view on the bigger picture of early mindreading capacities.

I only have some minor comments for the authors:

- Introduction, second paragraph: "a light and sound cue was emitted simultaneously" I think it should read "...cue were emitted..."
- Introduction, fourth paragraph: Dörrenberg et al. (2018) tested 24-month-olds, this study should be listed behind toddlers, not children. Schuwerk et al. (2016) tested 8-year-olds with and without ASC – I guess the authors refer to a different study: Schuwerk T, Prieswager B, Sodan B,

Perner J. 2018 The robustness and generalizability of findings on spontaneous false belief sensitivity: a replication attempt. *R.Soc.opensci.* 5:172273. <http://dx.doi.org/10.1098/rsos.172273>

- Introduction, fifth paragraph: The authors claim that there is only one published study meeting the criteria of a direct replication of the task reported by Southgate et al. in the 2007 paper. This statement seems to question the validity of all other previous replication studies. Yet, the authors do not go into detail for what reasons they discredit these studies. As far as I can see, many previous replication studies are at least close to direct replications with only minor deviations or used methods/age groups that resemble or lean on follow-up studies of this paradigm by the original authors. It is worth noting that also the current study differs in some ways from the original one (e.g., different dependent measure, data processing, exclusion criteria, ringing sound, etc.). To give readers the chance to make their own -unbiased- evaluation of the current situation, I would appreciate if the authors would provide some more information on how previous replication studies differed from the original study and explain why they think these differences would question their validity. For instance, some replication studies used a “four trial” familiarisation procedure (e.g., Dörrenberg et al. 2018, Kulke et al. 2018). Why should such a procedure have negative consequences on the results of these studies? In fact, the original authors themselves introduced the two extra familiarisation trials (that only show a short sequence of the agent reaching for a toy on the box) to the task in follow-up studies. Other replication studies used new stimulus material that they created in conformity with the original material (e.g., Kulke et al. 2018, Schuwert et al. 2018). Why should only the original footage work in eliciting belief-based action anticipation? And so forth.

Reviewer: 2

Comments to the Author(s)

The study is begging to be done. Over the last several years, there has been considerable discussion regarding the original findings from Southgate et al. (2007), whether these effects are robust, and what this means for research on early false-belief understanding. Yet, as the current authors note, no one has undertaken a *direct* replication, which makes this entire debate somewhat difficult to interpret. This direct replication conducted across 2 labs with a more robust sample size will thus make a valuable and important contribution to the field.

The study plan mirrors the original study quite closely, with only minor changes related to the eye-tracking equipment currently available. The authors plan to replicate the original analyses, while also conducting additional analyses to examine differences across lab, etc.

The preregistration on aspredicted.org is quite clear. However, the manuscript itself could be clarified in several places to help the reader understand how the current study is similar to/different from the original study that was conducted. The original paper was quite brief and did not discuss some criteria (such as exclusion) in great detail, so it would be helpful to provide additional detail in the current manuscript. Note all of these suggestions are intended to clarify the eventual manuscript. In my opinion, they do not affect the quality of the replication study itself, as all of these issues are clearly addressed in the online study plan.

- It is briefly mentioned on pg 8 that the original study coded gaze manually from a graphic representation of gaze generated in Clearview. The present study will instead use saccades/fixations as defined by the Eyelink software. It would be helpful to explain this difference earlier so that it is clear why two different AOIs are defined.

- Similarly, it would be helpful to state clearly in the manuscript whether the exclusion criteria were the same as in the original study. This is discussed in the supplementary material and preregistration, but it is somewhat unclear in the manuscript.

- In the preregistration, it states that the plan is to test 40 infants in each condition at each site (i.e. 160 infants total prior to applying the exclusion criteria). The manuscript does not lay this out as clearly, so it seems as though only 40 infants will be tested total. Presumably this will be clarified in the final manuscript when actual data are reported.

- The manuscript notes a discrepancy in the original paper regarding the analysis window. However, the nature of this discrepancy is somewhat unclear. From the preregistration, it seems like the duration from cue onset to the agent's reach (in familiarization) was 2750ms and the duration from cue *offset* to reach was 1750ms. But it is not clear which of these two windows as actually analyzed in the original paper - are the analyses in the original paper on the 2750ms window or the 1750ms window? In the current study, the authors plan to analyze both, so whichever it was the current paper will provide a direct replication of the original analyses. But it would be quite helpful to the field to clarify which was originally used, given that others have (or are currently) undertaken conceptual replications of this work.

Author's Response to Decision Letter for (RSOS-210190.R0)

See Appendix A.

Decision letter (RSOS-210190.R1)

Dear Dr Kampis

On behalf of the Editor, I am pleased to inform you that your Manuscript RSOS-210190.R1 entitled "A two-lab direct replication attempt of Southgate, Senju, & Csibra (2007)" has been accepted in principle for publication in Royal Society Open Science.

You may now progress to Stage 2 and complete the study as approved.

Please note that you must now register your approved protocol on the Open Science Framework (<https://osf.io/rr>), using the 'Submit your approved Registered Report' option and then the 'Registered Report Protocol Preregistration' option. Please use the Registered Report option even though your article is being accepted as a Stage 1 Replication. Further into the registration process, in the Journal Title field enter 'Royal Society Open Science (Replication article type, Results-Blind track)'. Please note that a time-stamped, independent registration of the protocol is mandatory under journal policy, and manuscripts that do not conform to this requirement cannot be considered at Stage 2. The protocol should be registered unchanged from its current approved state. Please include a URL to the protocol in your Stage 2 manuscript, and because you submitted via the Results-Blind track please note in the manuscript that the pre-registration was performed after data analysis (e.g. 'This article received results-blind in-principle acceptance (IPA) at Royal Society Open Science. Following IPA, the accepted Stage 1 version of the manuscript, not including results and discussion, was preregistered on the OSF (URL). This preregistration was performed after data analysis.')

Since you also preregistered the design before data analysis on AsPredicted, please also continue to list that existing pre-study registration at Stage 2 and be sure to distinguish it clearly from the registration of the Stage 1 manuscript above.

Following completion of your study, we invite you to resubmit your paper for peer review as a Stage 2 Replication. Please note that your manuscript can still be rejected for publication at Stage 2 if the Editors consider any of the following conditions to be met:

- The Introduction and methods deviated from the approved Stage 1 submission (required).
- The authors' conclusions were not considered justified given the data.

We encourage you to read the complete guidelines for authors concerning Stage 2 submissions at: <https://royalsocietypublishing.org/rsos/replication-studies#AuthorsGuidance>. Please especially note the requirements for data sharing and that withdrawing your manuscript will result in publication of a Withdrawn Registration.

Once again, thank you for submitting your manuscript to Royal Society Open Science and I look forward to receiving your Stage 2 submission. If you have any questions at all, please do not hesitate to get in touch. We look forward to hearing from you shortly with the anticipated submission date for your stage two manuscript.

Kind regards,
Professor Chris Chambers
Royal Society Open Science
openscience@royalsociety.org

on behalf of Professor Chris Chambers (Registered Reports Editor, Royal Society Open Science)
openscience@royalsociety.org

Author's Response to Decision Letter for (RSOS-210190.R1)

See Appendix B.

Decision letter (RSOS-210190.R2)

Dear Dr Kampis:

It is a pleasure to accept your manuscript entitled "A two-lab direct replication attempt of Southgate, Senju, & Csibra (2007)" in its current form for publication in Royal Society Open Science.

When you receive your proof, please can you ensure that the manuscript includes the URL to the accepted Stage 1 manuscript? The production office have been asked to include this, but you should check it has been included and - if it is not - you should provide it. Alternatively, you can email the editorial office with the details and they will be passed on to the production office.

Kind regards,
Professor Chris Chambers
Royal Society Open Science
openscience@royalsociety.org

Appendix A

Dear Editors,

Thank you for the feedback on the previous version of our manuscript. In light of the reviewers' comments we have made changes and clarifications at several points in the text, and hope to have addressed all points in a satisfactory manner. We outline our replies and indicate the changes we made in detail below.

Reviewer: 1

- Introduction, second paragraph: "a light and sound cue was emitted simultaneously" I think it should read "...cue were emitted..."

We corrected this phrasing.

- Introduction, fourth paragraph: Dörrenberg et al. (2018) tested 24-month-olds, this study should be listed behind toddlers, not children. Schuwerk et al. (2016) tested 8-year-olds with and without ASC – I guess the authors refer to a different study: Schuwerk T, Priewasser B, Sodian B, Perner J. 2018 The robustness and generalizability of findings on spontaneous false belief sensitivity: a replication attempt. *R.Soc.opensci.* 5:172273. <http://dx.doi.org/10.1098/rsos.172273>

We know merged references for "toddlers" and "children", and corrected Schuwerk et al. (2016) to (2018). We thank the reviewer for bringing this to our attention.

- Introduction, fifth paragraph: The authors claim that there is only one published study meeting the criteria of a direct replication of the task reported by Southgate et al. in the 2007 paper. This statement seems to question the validity of all other previous replication studies. Yet, the authors do not go into detail for what reasons they discredit these studies. As far as I can see, many previous replication studies are at least close to direct replications with only minor deviations or used methods/age groups that resemble or lean on follow-up studies of this paradigm by the original authors. It is worth noting that also the current study differs in some ways from the original one (e.g., different dependent measure, data processing, exclusion criteria, ringing sound, etc.). To give readers the chance to make their own - unbiased- evaluation of the current situation, I would appreciate if the authors would provide some more information on how previous replication studies differed from the original study and explain why they think these differences would question their validity. For instance, some replication studies used a "four trial" familiarisation procedure (e.g., Dörrenberg et al. 2018, Kulke et al. 2018). Why should such a procedure have negative consequences on the results of these studies? In fact, the original authors themselves introduced the two extra familiarisation trials (that only show a short sequence of the agent reaching for a toy on the box) to the task in follow-up studies. Other replication studies used new stimulus material that they created in conformity with the original material (e.g., Kulke et al. 2018, Schuwerk et al. 2018). Why should only the original footage work in eliciting belief-based action anticipation? And so forth.

The statement in no way questions the validity of the other replication studies, but it remains the case that most of the other replication attempts differed in some way from the original and so there always remains the possibility that deviation in results stems from deviation in protocol. It may well be that the changes are more or less important (i.e. four familiarization trials may have no impact, or they may increase boredom), but our point here is that as there is no prior direct replication, and that this will always leave some small doubt that the deviations caused the non-replications, we wanted to be absolutely sure by running a direct replication of the original paradigm.

Reviewer: 2

The preregistration on aspredicted.org is quite clear. However, the manuscript itself could be clarified in several places to help the reader understand how the current study is similar to/different from the original study that was conducted. The original paper was quite brief and did not discuss some criteria (such as exclusion) in great detail, so it would be helpful to provide additional detail in the current manuscript. Note all of these suggestions are intended to clarify the eventual manuscript. In my opinion, they do not affect the quality of the replication study itself, as all of these issues are clearly addressed in the online study plan.

- It is briefly mentioned on pg 8 that the original study coded gaze manually from a graphic representation of gaze generated in Clearview. The present study will instead use saccades/fixations as defined by the Eyelink software. It would be helpful to explain this difference earlier so that it is clear why two different AOIs are defined.

We have now expanded on this explanation at the beginning of the methods section (p. 4 /5) we outline differences between the original study and the present one, and in order to keep this section easy to follow we also kept the references to later points and to the SI:

“(3) the first look was operationalised as fixation on either a window or a box to approximate the original manual coding of saccades to windows (see Dependent measures for details) [...]

[...] (7) the data-processing pipeline was implemented differently: in the original saccades and fixations were coded manual from a graphic representation of gaze generated in Clearview software, the present study used numerical data on saccades and fixations generated by the Eyelink online parser (see Dependent measures below and Supplementary Information).”

- Similarly, it would be helpful to state clearly in the manuscript whether the exclusion criteria were the same as in the original study. This is discussed in the supplementary material and preregistration, but it is somewhat unclear in the manuscript.

We now include in the Participants section a description taken from the original study regarding their exclusion criteria, as well as a statement that the present study’s exclusions were set to closely approximate these and define them in more detail (p. 5).

“All exclusion criteria were preregistered, and were matched as closely as possible to the original study (for details see pre-registration and SI). In the original, participants were excluded if they did not correctly anticipate the opening of the right-hand window on the second familiarization trial, looked away at the crucial moment on the test trial, did not look at either window on the test trial, or the eyetracker could not be calibrated for them. We specified analogous criteria in more detail in our pre-registration. We intended to ensure that infants in the included sample were attending to key events in familiarisation trials and thus established a criterion for minimum looking during pre-defined events (see below). Note that the original paper did not report any exclusions resulting specifically from infants looking away during these events. In the supplement we also analysed the data including infants who did not meet our criteria of attending to key events in familiarisation and test trials.”

- In the preregistration, it states that the plan is to test 40 infants in each condition at each site (i.e. 160 infants total prior to applying the exclusion criteria). The manuscript does not lay this out as clearly, so

it seems as though only 40 infants will be tested total. Presumably this will be clarified in the final manuscript when actual data are reported.

Indeed, this may have been ambiguous in the manuscript. We have now added to the beginning of the Participants section (p. 5) the following sentence:

“We thus set the sample size at 40 infants tested in each condition at each site (i.e., 160 infants total prior to applying the exclusion criteria).”

- The manuscript notes a discrepancy in the original paper regarding the analysis window. However, the nature of this discrepancy is somewhat unclear. From the preregistration, it seems like the duration from cue onset to the agent's reach (in familiarization) was 2750ms and the duration from cue *offset* to reach was 1750ms. But it is not clear which of these two windows was actually analyzed in the original paper - are the analyses in the original paper on the 2750ms window or the 1750ms window? In the current study, the authors plan to analyze both, so whichever it was the current paper will provide a direct replication of the original analyses. But it would be quite helpful to the field to clarify which was originally used, given that others have (or are currently) undertaken conceptual replications of this work.

We write on p. 7 in Dependent measures (par.2):

“The time window of interest was defined as the 2750 ms interval starting from the onset of the acoustic and visual cues that indicated the agent’s impending action. In the original paper, the time window of analysis is described as 1750 ms from the cue onset and it was said to correspond to the time-lag between the cue onset and window opening in familiarization. In fact this time-lag lasts 2750 ms. After inspecting the original stimuli and data as well as consulting among the authors, we concluded that the length of 1750 ms was reported incorrectly in the original methods and we corrected it to 2750 ms for the current replication. We do, however, also report analyses for the time window lasting 1750 ms from the cue onset (see SI).”

Thus, while in the original paper the analysis window was specified as 1750 ms from the cue onset. But since it was also said to correspond to the time-lag between the cue onset and window opening in familiarization, these two pieces of information do not align (it was either 1750 ms from cue onset, or the same time-lag as in familiarization between cue onset and window opening in familiarization). Thus, we concluded that most likely the analysis window in the original paper was in fact 2750ms from cue onset and may have been a typo in the description. However, as it is not possible to fully reconstruct the original analyses, and it may have been that the time window of analysis in test was in fact 1750 ms from cue onset (however not corresponding to the above described time-lag in familiarization), the current paper reports both time windows, as indicated in the above paragraph.

We thank both reviewers for their comments on the manuscript and believe the paper has become clearer in light of these clarifications.

*We look forward to hearing from you,
Sincerely,*

The authors

Appendix B

UNIVERSITY OF COPENHAGEN
DEPARTMENT OF PSYCHOLOGY

Dear Editors,

Please find enclosed the abstract of our manuscript entitled “*A two-lab direct replication attempt of Southgate, Senju, & Csibra (2007)*” which we submit as a Stage 2 results-blind track Replication study to Royal Society Open Science. The article received in-principle acceptance in the journal on March 8, 2021. The attached submission is the completed Stage-2 manuscript of this project.

We have pre-registered the Stage 1 accepted manuscript on the OSF (https://osf.io/bsf2r?view_only=0a90aeddc40543fb85f7ef8e8e8f53cf) and archived all anonymised data, code and digital materials (<https://osf.io/86wq2/>).

Thank you for considering our manuscript for publication in Royal Society Open Science.

Sincerely,

Dora Kampis¹, Petra Kármán, Gergely Csibra, Victoria Southgate, & Mikołaj Hernik

¹Department of Psychology
University of Copenhagen
Øster Farimagsgade 2A, 1353 Copenhagen, Denmark
e-mail: dk@psy.ku.dk